# DISCRETE INFOMAX CODES FOR META-LEARNING

## ABSTRACT

This paper analyzes how generalization works in meta-learning. Our core contribution is an information-theoretic generalization bound for meta-learning, which identifies the expressivity of the task-specific learner as the key factor that makes generalization to new datasets difficult. Taking inspiration from our bound, we present Discrete InfoMax Codes (DIMCO), a novel meta-learning model that trains a stochastic encoder to output discrete codes. Experiments show that DIMCO requires less memory and less time for similar performance to previous metric learning methods and that our method generalizes particularly well in a challenging small-data setting.

## 1 INTRODUCTION

Generalizing to unseen data is a problem of vital importance in machine learning. Many deep meta-learning methods optimize for generalization by directly minimizing the loss of held-out validation data. Recent works have used this framework to achieve impressive feats such as learning to classify using one labeled image per class (Snell et al., 2017; Finn et al., 2017), learning unsupervised update rules that generalize to different domains (Metz et al., 2018), and accelerating training procedures that are millions of steps long (Flennerhag et al., 2018). However, the meta-learning setup introduces a new overfitting problem: *the model may overfit to the distribution of tasks seen during training*. In other words, the meta-learning framework decreases task-specific overfitting at the cost of introducing task-wise overfitting.

The primary aim of this work is to elucidate this tradeoff between task-specific and task-wise overfitting in meta-learning. Specifically, we use tools from information theory to bound the overall generalization gap of a meta-learner. Analogously to how generalization bounds for standard learning algorithms reveal the role of dataset size in generalizing to new *datapoints*, our bound reveals the roles of both dataset size ($m$) and number of datasets ($n$) in generalizing to new *tasks*. The specific form of our generalization gap suggests that meta-learning models can be implicitly regularized by constructing them to express each datapoint with a small number of bits.

To this end, we propose **D**iscrete **I**nfo**M**ax **CO**des (DIMCO), a deep neural network model that outputs a *discrete representation* of each datapoint. Note that a continuous representation $\tilde{x} \in \mathbb{R}^n$ (e.g. Snell et al. (2017)) uses $32n$ bits per datapoint. This is wildly inefficient in terms of bit-efficiency: our experiments in Section 5 show that DIMCO's discrete representation requires roughly $10\times$ less bits per datapoint to achieve similar performance compared to continuous methods. DIMCO generalizes well to novel datasets because its learning objective encourages the model to compactly use all of its degrees of freedom, thus enabling it to effectively compare datapoints while only requiring a small number of bits per datapoint.

Our specific contributions are:

1. Derive a generalization bound for meta-learning that makes the tradeoff between task-specific and task-wise overfitting concrete.

2. Propose DIMCO, a neural network model that is designed to have a low value of a specific term in our generalization bound.

3. Empirically demonstrate that DIMCO generalizes better than previous meta-learning methods when trained with small datasets, and that it is more memory- and time-efficient compared to previous image retrieval methods.

This paper is organized as follows. We detail our problem setup and derive a generalization bound for meta-learning in Section 2. Taking motivation from our bound, we propose our meta-learning model (DIMCO) in Section 3. We put our analysis in the context of previous work in Section 4. Notably, we suggest that certain previous meta-learning methods may have benefitted from implicit regularization. We present experiments in Section 5 and conclude the paper with a discussion about limitations and future directions of our approach in Section 6.

## 2 A GENERALIZATION BOUND FOR META-LEARNING

Throughout this section, we denote data, model outputs, and labels as $\mathbf{x}$, $\widetilde{\mathbf{x}}$, and $\mathbf{y}$, respectively. We use capital symbols $X$, $\widetilde{X}$, $Y$ to denote the random variables corresponding to $\mathbf{x}$, $\widetilde{\mathbf{x}}$, $\mathbf{y}$.

The (discrete) *Mutual Information* between two random variables $X_1, X_2$ is defined as

$$I(X_1; X_2) = H(X_1) - H(X_1|X_2) = H(X_2) - H(X_2|X_1) \tag{1}$$

$I(X_1; X_2)$ is a symmetric quantity which measures the amount of information shared between $X_1$ and $X_2$. It has its lowest value 0 when $X_1$ and $X_2$ are independent and increases with the correlation between $X_1$ and $X_2$. We refer the reader to (Cover & Thomas, 2012) for further exposition.

### 2.1 PROBLEM SETUP

We begin by describing our meta-learning problem setup. Define a task $T$ to be a distribution over $\mathcal{Z} = \mathcal{X} \times \mathcal{Y}$. Let tasks $T^1, \ldots, T^n$ be sampled i.i.d. from a distribution of tasks $\tau$. Associated with each task $T$, we define a dataset $D_T = z_T^1, \ldots, z_T^m = (x_T^1, y_T^1), \ldots, (x_T^m, y_T^m)$ which is a set of $m$ i.i.d. samples from the data distribution ($z_T^j \sim T$).

We consider models with parameters $\theta$ that map datapoints $X$ to representations $\widetilde{X}(X, \theta)$. Our objective is the expecteation of the negative mutual information between representations an labels across all tasks:

$$\mathcal{L}(\tau, \theta) = -\mathbb{E}_{T \sim \tau} \left[ I(\widetilde{X}(X_T, \theta); Y_T) \right]. \tag{2}$$

This objective is closely related to both previous loss functions and evaluation metrics in setups that involve testing on unseen classes (e.g. few-shot classification, image retrieval). We show that previous loss functions can be seen as approximations to (2) in Appendix A, and that mutual information is strongly correlated with metrics such as few-shot accuracy and Recall@1 in Section 5.1.

An important difference between (2) and previous objectives for few-shot classification is that we do not split each task into support/query (also called train/test) sets. This property bypasses the pesky issue of using batch normalization (BN) during meta-training. As Nichol & Schulman (2018) points out, BN can leak information from the support set to the query set. This issue is not a problem in our setup since we do not assume a seperate "query set". Additionally, not using support/query splits enables meta-learning with tasks consisting of one image per class, as we demonstrate in Section 5.4. Note that the standard meta-learning setup cannot learn from such tasks since it requires at least two images per class (one for support, one for query) to compute the loss function. Overall, our model demonstrates that the commonly used construction of a held-out test set within each task is not strictly necessary for meta-learning.

### 2.2 GENERALIZATION BOUND

We bound the difference between expected loss and empirical loss:

**Theorem 1.** *Let $\tau, n, m, X, \widetilde{X}, Y, \theta, \mathcal{L}$ be defined as above. Let $d_\Theta$ be the VC dimension of the encoder $\widetilde{X}(\cdot)$. Let $\hat{I}(\widetilde{X}(X_T, \theta); Y_T)$ be the empirical estimate of the mutual information using finite dataset $D_T$, and define empirical loss as*

$$\hat{\mathcal{L}}(T^{1:n}, \theta) = -\frac{1}{n} \sum_{i=1}^{n} \hat{I}(\widetilde{X}(X_{T^i}, \theta); Y_{T^i}). \tag{3}$$

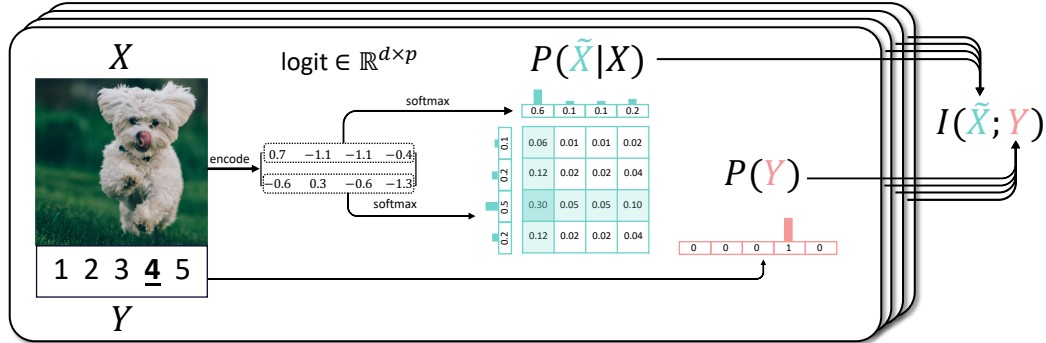

Figure 1: A graphical overview of **D**iscrete **I**nfo**M**ax **CO**des (DIMCO). A dataset $D$ consists of pairs of images $X$ and labels $Y$. DIMCO is a stochastic encoder that maps each image $X$ to a distribution of discrete codes $p(\widetilde{X}|X)$. Each discrete code $\widetilde{\mathbf{x}} \sim p(\widetilde{X}|X)$ is a $p$-way code of length $d$. If $p = 4, d = 2$ (as in the diagram), each code consists of 2 symbols and each symbol is $\in \{1, 2, 3, 4\}$. Inside the $4 \times 4$ grid that represents the $p^d = 16$ possible codes, the most likely row and column are colored. The most likely code in the diagram is $(1, 2)$ with probability $30\%$. DIMCO is optimized by maximizing the mutual information between the discrete code and the label within each batch.

*The following inequality holds with high probability:*

$$\mathcal{L}(\tau, \theta) - \hat{\mathcal{L}}(T^{1:n}, \theta) \leq O\left(\sqrt{\frac{d_\Theta}{n} \log \frac{n}{d_\Theta}}\right) + O\left(\frac{|\widetilde{X}| \log(m)}{\sqrt{m}}\right) + O\left(\frac{|\widetilde{X}||Y|}{m}\right) \qquad (4)$$

*Proof.* We use standard Chernoff bounds along with a finite sample bound for mutual information from Shamir et al. (2010); see Appendix B. □

The generalization gap has three terms, two of which decrease as $m$ increases, and the other decreases as $n$ increases. Typically for few-shot learning, $n$ is very large while $m$ is small: the typical miniImagenet 5-way 1-shot setup has $n > 10^{10}$ and $m = 5$. We therefore claim that **the latter two terms are the main difficulties for generalizing to new tasks**. We see from Theorem 1 that these terms can be reduced by using small $|\widetilde{X}|$. Therefore, in the context of meta-learning, using short representations (i.e. small $|\widetilde{X}|$) can compensate for having a small train set (i.e. small $m$).

Meta-learning is typically formulated as performing two levels of learning: task-general learning and task-specific learning. Theorem 1 implies that in the tasks considered in recent literature, task-general learning should have almost no generalization gap, making task-specific learning the main source of (meta-)overfitting. This can be problematic since in many works, the task-specific learning algorithm is usually just a byproduct of whatever clever meta-learning loss was proposed. Our theorem suggests that we should pay more attention to directly regularizing this task-specific learner, and our DIMCO model, described in the next section, can be seen as a minimal working example in this direction.

## 3 DISCRETE INFOMAX CODES (DIMCO)

We now present our model, **D**iscrete **I**nfo**M**ax **CO**des (DIMCO). Motivated by Section 2, DIMCO produces a short discrete code $\widetilde{X}$ and is trained by maximizing mutual information $I(\widetilde{X}; Y)$. Figure 1 graphically shows the overall structure of DIMCO.

### 3.1 FACTORIZED DISCRETE CODES

We propose a factorized discrete representation scheme which enables us to represent discrete distributions with exponentially fewer parameters compared to listing the probability of each event. We represent each event as the product of $d$ independent events, each of which consists of $p$ different possibilities. We thus have $p^d$ events in total, but only require $pd$ parameters to represent the

probability of each event. Binary codes can be viewed as a special case of this scheme where $p = 2$. This factorization trick allows us to consider representations of size $|\widetilde{X}| = 64^{256}$ (Section 5). This representation has the advantage of requiring only $d \log_2 p$ bits per datapoint, whereas a $D$-dimensional continuous vector embedding requires $32d$ bits (assuming 32-bit floats).

### 3.2 MODEL

Recall that we represent a given image using $d$ independent discrete distributions, each of which has $p$ possibilities. First, a (convolutional) neural network $\text{enc}(\cdot)$ takes image $X$ as input and outputs a vector of length $dp$, which we reshape into a matrix of size $d \times p$:

$$\text{enc}(X) = \begin{bmatrix} l_{11} & l_{12} & l_{13} & \ldots & l_{1p} \\ l_{21} & l_{22} & l_{23} & \ldots & l_{2p} \\ \vdots & \vdots & \vdots & \ddots & \vdots \\ l_{d1} & l_{d2} & l_{d3} & \ldots & l_{dp} \end{bmatrix} \tag{5}$$

Each row of this matrix represents the logits of a discrete distribution. We apply the softmax function to each row to get probabilities.

$$\text{softmax}(l_{i1}, \ldots, l_{ip}) = p_{i1}, \ldots, p_{ip} \tag{6}$$

The $i$th codeword is sampled according to the categorical distribution following these probabilites:

$$\widetilde{x}_i \sim \text{Cat}(p_{i1}, \ldots, p_{ip}). \tag{7}$$

We simply concatenate each $\widetilde{x}_i$ to obtain the representation: $\widetilde{\mathbf{x}} = (\widetilde{x}_1, \ldots, \widetilde{x}_d)$.

### 3.3 TRAINING

Recall that $\widetilde{\mathbf{x}}$ is a discrete random variable and $\widetilde{X}$ is its distribution. Instead of sampling $\widetilde{\mathbf{x}} \sim \widetilde{X}$, we directly use $\widetilde{X}$ to compute the objective:

$$I(\widetilde{X}; Y) \quad = \quad H(\widetilde{X}) - H(\widetilde{X}|Y). \tag{8}$$

The first term, $H(\widetilde{X})$, can be calculated by taking the average of all probabilities and computing the entropy:

$$H(\widetilde{X}) = \sum_{i=1}^{d} H(\widetilde{X}_i) = \sum_{i=1}^{d} H\left( \text{Cat}\left( \frac{\sum_{j=1}^{m} p_{i1}^j}{m}, \frac{\sum_{j=1}^{m} p_{i2}^j}{m}, \ldots, \frac{\sum_{j=1}^{m} p_{ip}^j}{m} \right) \right) \tag{9}$$

The second term is $H(\widetilde{X}|Y) = \sum_{k=1}^{c} p(Y = k) H(\widetilde{X}|Y = k)$ where $c$ is the number of classes. The marginal probability of Y ($p(Y = k)$) is the frequency of $k$ in $\{\mathbf{y}^1, \ldots, \mathbf{y}^m\}$. $H(\widetilde{X}|Y = k)$ can be obtained by computing (9) using only $\mathbf{x}^j$ for which $\mathbf{y}^j = k$.

Though we have motivated the use of $I(\widetilde{X}; Y)$ as a loss function throughout this paper, we provide yet another perspective using the decomposition in (8). Minimizing $H(\widetilde{X}|Y)$ encourages discriminatory behavior. This term encourages the average embedding of each class to be as concentrated as possible. Maximizing $H(\widetilde{X})$ incentivizes the model to overall use all possible values of $\widetilde{X}$.

We emphasize that such closed-form computation of $I(\widetilde{X}; Y)$ is only possible because we are using discrete codes.

### 3.4 EVALUATION

We map all images to their probabilities $p_{ij}$ via (5) and (6) for all $i = 1, \ldots, d$ and $j = 1, \ldots, p$. We map each training image to its most likely code:

$$\widetilde{\mathbf{x}} = \left( \overbrace{\arg\max_{j} p_{1j}}^{\widetilde{x}_1}, \ \overbrace{\arg\max_{j} p_{2j}}^{\widetilde{x}_2}, \ \cdots, \ \overbrace{\arg\max_{j} p_{dj}}^{\widetilde{x}_d} \right) \tag{10}$$

Fix a train image and a test image, and let $\widetilde{\mathbf{x}}$ be the most likely code for the train image. The similarity between train image and test image is measured by the probability of the test image producing $\widetilde{\mathbf{x}}$. This amounts to computing the product [1] of the test image's probabilites using $\widetilde{\mathbf{x}}_i$ for each $i = 1, \ldots, d$:

$$\prod_{i=1}^{d} p_{\widetilde{\mathbf{x}}_i i}. \tag{11}$$

We use this as a similarity metric for both few-shot classification and image retrieval. We perform few-shot classification by computing the most likely code for each class via (10) and classifying each test image by choosing the class that has highest value of (11). We similarly perform image retrieval by mapping each support image to its most likely code (10) and for each query image retrieving the support image that has highest (11).

## 4   RELATED WORK

**Regularizing Meta-Learners**   The ability to generalize to novel datasets is critical in meta-learning benchmarks, and even more so in benchmarks such as *Meta-Dataset* (Triantafillou et al., 2019), where a model is tested on datasets from an unseen domain. To the best of our knowledge, no works have proposed an explicit regularizer for generalizing to new tasks.

Our analysis suggests that the success of some previous meta-learning methods can be attributed to being implicitly regularized by reducing the expressive power reducing task-specific learning. The following works have reported benefits from reducing the number of such task-specific parameters: Lee & Choi (2018) learns a subset of the full network to alter during task-specific learning, Rusu et al. (2018) explicitly represents each task with a low-dimensional latent space, and Zintgraf et al. (2018) alters only a pre-specified subset of the full network during task-specific learning. It may be surprising at first that these methods achieve higher accuracy than vanila MAML (Finn et al., 2017), which is more expressive since it alters all parameters during task-specific learning. Kim et al. (2018) also reports better meta-generalization through approximate variational inference with respect to a learned prior, which can be seen as restricting the search space of the task-specific learner. We showed through Theorem 1 that restricting inner-loop expressivity reduces the generalization gap; this provides theoretical understanding to this consensus that meta-learning models with simple task-specific learners generalize to new tasks more easily.

**Information Bottleneck**   Theorem 1 is close in spirit to the information bottleneck principle (Tishby et al., 2000; Tishby & Zaslavsky, 2015; Shwartz-Ziv & Tishby, 2017). This principle states that to generalize, one should maximize $I(\widetilde{X}; Y)$ while simultaneously minimizing $I(\widetilde{X}; X)$. Likewise, our objective (2) is $I(\widetilde{X}; Y)$ while our bound (22) suggests that the representation capacity $|\widetilde{X}|$ should be low for generalization. Also related is the deterministic information bottleneck (Strouse & Schwab, 2017) which extends the information bottleneck by minimizing $H(\widetilde{X})$ rather than $I(\widetilde{X}; X)$. These three approaches to generalization are related via the chain of inequalities $I(\widetilde{X}; X) \leq H(\widetilde{X}) \leq \log |\widetilde{X}|$, which is tight when $\widetilde{X}$ is an efficient code.

**Representation Learning**   Previous works have applied information-theoretic principles to analyze the objective of VAEs (Alemi et al., 2017; Chen et al., 2018), derive an objective for GANs to learn disentangled features (Chen et al., 2016), and to directly learn representations (Alemi et al., 2016; Hjelm et al., 2018; Oord et al., 2018; Grover & Ermon, 2018; Choi et al., 2019). Our work can also be viewed as an information-theoretic representation learning method, but we assume a supervised meta-learning setup and our main focus is the meta-generalization problem.

**Discrete Representations**   Discrete representations have been thoroughly studied in the context of information theory (Shannon, 1948). Recent deep learning methods directly learn discrete representations, by learning variational autoencoders with discrete latent variables (Rolfe, 2016; van den Oord et al., 2017; Razavi et al., 2019) or maximizing the mutual information between representation and

---

[1] In practice, we add log probabilities instead of multiplying probabilities for numerical stability.

data (Hu et al., 2017). DIMCO is related to but differs from these works as it assumes a supervised meta-learning setting and performs infomax using *labels* instead of data.

Similarly to our setup, Jeong & Song (2018) uses labels to learn a binary hash code. Their focus is on the speedup gained by using sparse codes, whereas DIMCO learns a dense discrete code to generalize better. Additionally, their method solves a minimum cost flow problem within each batch to find the locally optimal code, whereas DIMCO is able to directly compute its loss function.

**Factorized Representations** The idea of using factorized representations has appeared the contexts of quantizing a continuous input (Jegou et al., 2011), memory-efficient clustering (Norouzi & Fleet, 2013), and constructing an expressive attention mechanism using few parameters (Vaswani et al., 2017). Likewise, DIMCO factorizes its discrete representatations to increase its representation power.

**Metric Learning** The structure and loss function of DIMCO is closely related to embedding-based meta-learning Vinyals et al. (2016); Snell et al. (2017); Oreshkin et al. (2018) and image retrieval Hoffer & Ailon (2015); Sohn (2016); Wu et al. (2017); Duan et al. (2018) methods. We show in Appendix A that the loss functions of these methods can be seen as approximation to the mutual information ($I(\widetilde{X}; Y)$). While all of these previous methods require a support/query split within each task, DIMCO simply optimizes an information-theoretic quantity of each batch, removing the need for such structured batch construction.

## 5 EXPERIMENTS

We use the miniImageNet (Ravi & Larochelle, 2016) and CUB200 (Wah et al., 2011) datasets with standard splits for both in our experiments. The miniImageNet dataset is a subset of the Imagenet (Krizhevsky et al., 2012) dataset that was made for few-shot classification. It consists of 100 classes each containing 600 images of size $84 \times 84$. The classes are split into 64 training, 16 validation, and 24 test classes. The Caltech-UCSD Birds-200-2011 (CUB200) dataset consists of 11788 images of birds from 200 classes. The classes are split into 100 training and 100 test classes.

We use two different CNN backbones for our experiments: the 4-layer convnet commonly used for meta-learning (Finn et al., 2017; Sung et al., 2018; Liu et al., 2018), and the Inception network (Szegedy et al., 2015) with batch normalization (Ioffe & Szegedy, 2015) which is commonly used for deep image retrieval (Sohn, 2016; Movshovitz-Attias et al., 2017; Wu et al., 2017). We randomly initialize weights for the 4-layer convnet and use pretrained weights for the Inception network

### 5.1 CORRELATION OF METRICS

This experiment empirically verifies whether mutual information $I(\widetilde{X}; Y)$ is a reasonable metric for quality of representation. We trained DIMCO on the miniImagenet dataset with $p = d = 64$ for 20 epochs, for 8 independent runs. We plot the pairwise correlations between five different metrics ( $(5, 10, 20)$-way 1-shot accuracy, Recall@1, and $I(\widetilde{X}; Y)$ ) in Figure 2. We see that all five metrics are very strongly correlated. We observed similar trends when training with with previously proposed loss functions: we visualize these results in Figure 5 of the appendix due to space constraints. Alongside this empirical evidence, we prove in Appendix A that previously used loss functions for few-shot classification and image retrieval are approximations to $I(\widetilde{X}; Y)$.

### 5.2 WHAT DOES EACH CODE REPRESENT?

We inspected what each code represents in a DIMCO model ($d = 16$, $p = 64$) trained on the miniImagenet dataset. Recall that each image produces a $d \times p$ probability matrix (5, 6). For each of these $dp$ entries, we plotted the *top* 10 *images in the test set* that assigned highest probability to that entry. We show images corresponding to four such codes in Figure 2 (right) and more in Figure 6 of the appendix.

We see that DIMCO learns a distributed representation. For example, the top code in Figure 2 represents the high-level concept of a furry animal and the shown 10 images span 4 different classes. On the other hand, the bottom code seems to focus on the color of the background. By aggregating

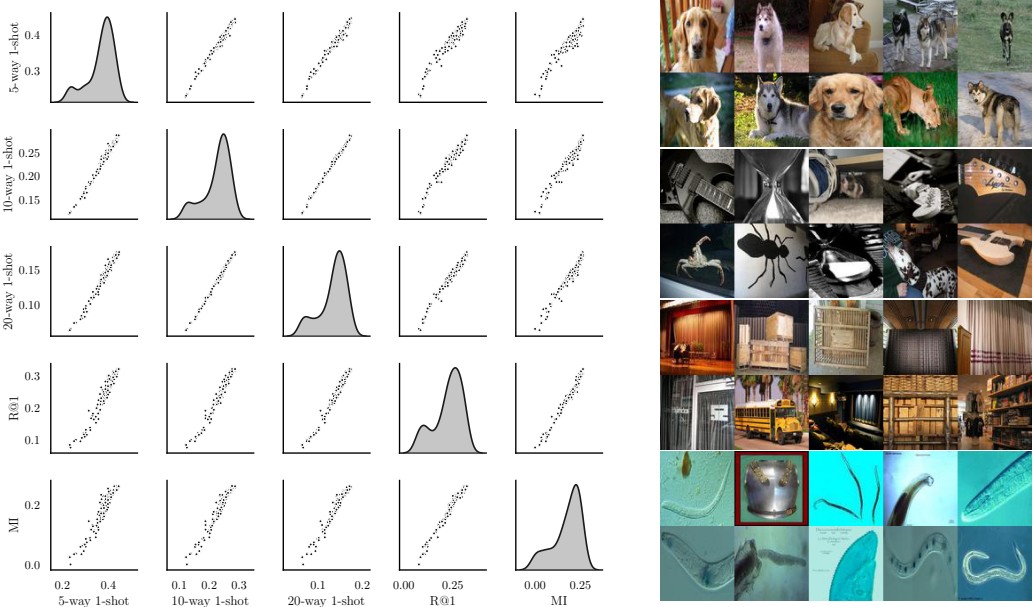

Figure 2: Left: Pairwise correlation between MI $= I(X; \widetilde{X})$ and previous metrics. Right: Visualization of codes of a small trained DIMCO model ($d = 16$, $p = 64$); we show the top 10 test set images that assign highest probability to a specific code, for 4 different codes.

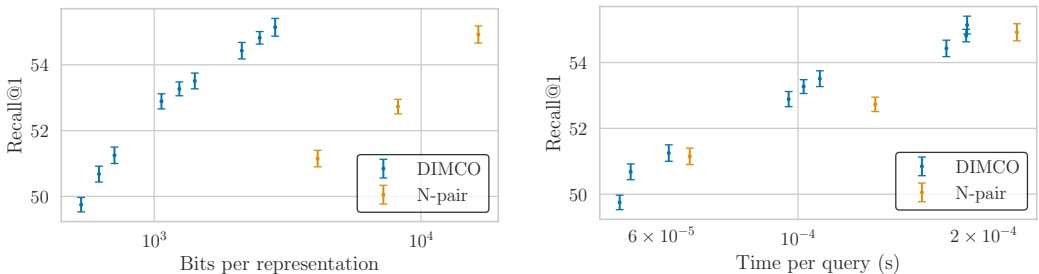

Figure 3: Image retrieval performance of DIMCO and N-pair loss on the CUB-200 dataset. The y-axis for both figures are the Recall@1 metric, and error bars reflect standard deviation computed from $n = 5$ runs per configuration. The x-axes represent (left) bits required to store one representation and (right) seconds required to perform retrieval for one query. Both x-axes are log-scale.

such complementary features in each of its $d$ codewords, DIMCO is able to classify images that belong to previously unseen classes.

## 5.3    TIME- AND MEMORY-EFFICIENT IMAGE RETRIEVAL

We conducted an image retrieval experiment using the CUB200 dataset, and used multiclass N-pair loss (Sohn, 2016) as a baseline. As is standard for image retrieval, we use the Inception network as specified in the beginning of this section. Using the same backbone, we trained DIMCO with $(p, d) \in \{64, 128, 256\} \times \{128, 256, 512\}$ and multiclass N-pair with embedding dimension $\in \{128, 256, 512\}$. We measured the time per query for each method on a single Tesla P40 GPU by averaging the time required for 10000 batches of queries of size 32.

Results in Figure 3 show that the compact code of DIMCO takes roughly an order of magnitude less memory for similar performance to N-pair loss, and requires less query time as well. This experiment also demonstrates that discrete representations can match the performance of modern methods that use continuous embeddings on this relatively large-scale task. We additionally note that DIMCO is

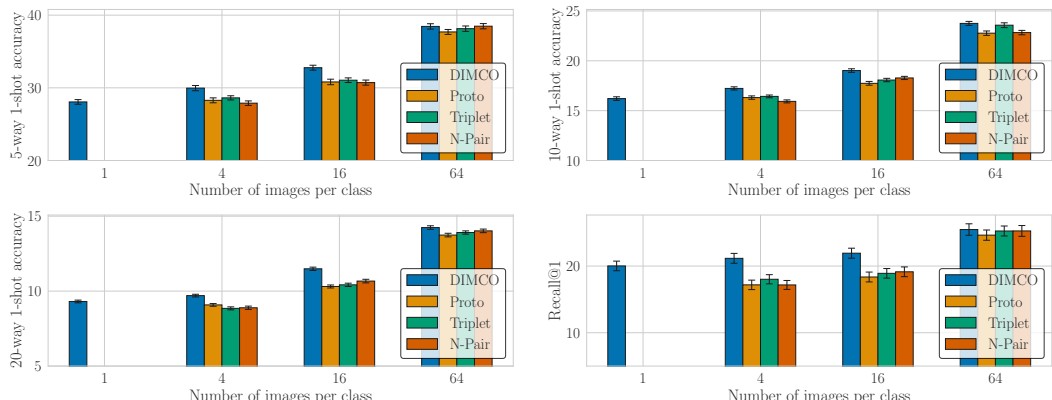

Figure 4: Performance of methods trained using subsets of miniImageNet of varying size. The lowermost y axis value for each metric corresponds to the expected performance of random guessing.

able to train using large backbones without significantly overfitting, whereas experiments reported in Mishra et al. (2017) indicate that MAML (Finn et al., 2017) overfits tremendously when using a deeper backbone.

### 5.4 GENERALIZATION TO NEW TASKS

This experiment measures how well DIMCO can generalize to new datasets *after training with a small number of datasets*. This challenging experimental setup measures how much generalizable information the model can extract from a limited set of datasets; it can be seen as the meta-learning analogue of measuring the performance of a classifier trained with a small dataset.

We trained each model using $\{1, 4, 16, 64\}$ samples from each training class in the miniImageNet dataset. For example, by using 4 samples, we are reducing the full train split of (64 classes $\times$ 600 images per class) into (64 classes $\times$ 4 images per class). We compare against three baseline methods: Triplet Nets(Hoffer & Ailon, 2015), multiclass N-pair loss(Sohn, 2016), and ProtoNets(Snell et al., 2017). We report the average and standard deviation of the top 5 results of a random hyperparameter search (see appendix for details). We show $\{5, 10, 20\}$-way 1-shot accuracies and Recall@1 of the test set in Figure 4.

First note that DIMCO is the only method that can train using a dataset consisting of 1 example per class. This is because other methods require at least one train and test example per class within each batch, while DIMCO requires no such train/test (also called support/query) split and simply maximizes the mutual information within a batch. Furthermore, Figure 4 shows that DIMCO learns much more effectively when the number of examples per class is low; this is because DIMCO uses fewer bits to describe each datapoint, which lowers its generalization gap (24) when applying to novel datasets.

## 6 CONCLUSION AND DISCUSSION

**Summary**    We derived a generalization bound for meta-learning using information-theoretic principles. Building on our bound, we proposed DIMCO, a model that learns a discrete representation of data by maximizing its mutual information with the label. DIMCO had benefits in time, memory, and generalization in our experiments.

**Towards Explicit Meta-Regularization**    In Section 4, we have suggested with analogy to Theorem 1 that the benefits of some previous meta-learning methods can be attributed to implicitly being regularized by reducing the expressivity of their task-specific learners. While DIMCO has demonstrated better generalization by reducing this expressivity via hyperparameters $(d, p)$, this is only applicable to DIMCO's specific setup of few-shot classification through mutual information maximization. In future work, we would like to explore explicit meta-regularization schemes that

can be applied to other problems (regression, reinforcement learning etc.) and algorithms (MAML, Neural Process etc.).

**Meta-Learning Without Splits**    Our meta-learning problem setup in Section 2.1 does not assume a support/query (also called train/test) split within each dataset. Along with showing that the traditional support/query split is not strictly necessary, we demonstrated in Section 5.4 that removing it has the benefit of enabling meta-learning in datasets having one image per class. We believe future work could benefit from further exploring this space of meta-learning algorithms that use datasets without splits.

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

## A    PREVIOUS LOSS FUNCTIONS ARE APPROXIMATIONS TO MUTUAL INFORMATION

**Cross-entropy Loss**    The cross-entropy loss has directly been used for few-shot classification (Vinyals et al., 2016; Snell et al., 2017).

Let $q(\mathbf{y}|\widetilde{\mathbf{x}};\phi)$ be a parameterized prediction of $\mathbf{y}$ given $\widetilde{\mathbf{x}}$, which tries to approximate the true conditional distribution $q(\mathbf{y}|\widetilde{\mathbf{x}})$. Typically in a classification network, $\phi$ is the parameters of a learned projection matrix and $q(\cdot)$ is the final linear layer. The expected cross-entropy loss can be written as

$$\text{xent}(Y,\widetilde{X}) = \mathbb{E}_{\mathbf{y}\sim Y,\widetilde{\mathbf{x}}\sim\widetilde{X}}\left[-\log q(\mathbf{y}|\widetilde{\mathbf{x}},\phi)\right]. \tag{12}$$

Assuming that the approximate distribution $q(\cdot)$ is sufficiently close to $p(\mathbf{y}|\widetilde{\mathbf{x}})$, minimizing (12) can be seen as

$$\arg\min \text{xent}(Y,\widetilde{X}) \quad \approx \quad \arg\min \mathbb{E}_{\mathbf{y}\sim Y,\widetilde{\mathbf{x}}\sim\widetilde{X}}\left[-\log p(\mathbf{y}|\widetilde{\mathbf{x}})\right] \tag{13}$$

$$= \quad \arg\min H(Y|\widetilde{X}) = \arg\max I(\widetilde{X};Y), \tag{14}$$

where the last equality uses the fact that $H(Y)$ is independent of model parameters. Therefore, cross-entropy minimization is approximate maximization of the mutual information between representation $\widetilde{X}$ and labels $Y$.

The approximation is that we parameterized $q(\mathbf{y}|\widetilde{\mathbf{x}};\phi)$ as a linear projection. This structure cannot generalize to new classes because the parameters $\phi$ are specific to the labels $\mathbf{y}$ seen during training. For a model to generalize to unseen classes, one must amortize the learning of this approximate conditional distribution. (Vinyals et al., 2016; Snell et al., 2017) sidestepped this issue by using the embeddings for each class as $\phi$.

**Triplet Loss**    The Triplet loss (Hoffer & Ailon, 2015) is defined as

$$\mathcal{L}_{\text{triplet}} = \left\|\widetilde{\mathbf{x}}_q - \widetilde{\mathbf{x}}_p\right\|_2^2 - \left\|\widetilde{\mathbf{x}}_q - \widetilde{\mathbf{x}}_n\right\|_2^2, \tag{15}$$

where $\widetilde{\mathbf{x}}_q,\widetilde{\mathbf{x}}_p,\widetilde{\mathbf{x}}_n \in \mathbb{R}^d$ are the embedding vectors of query, positive, and negative images. Let $\mathbf{y}_q$ denote the label of the query data. Recall that the pdf function of a unit Gaussian is $\log N(\widetilde{\mathbf{x}}|\mu,1) = -c_1 - c_2\|\widetilde{\mathbf{x}} - \mu\|_2^2$, where $c_1, c_2$ are constants. Let $p_p(\widetilde{\mathbf{x}}) = N(\widetilde{\mathbf{x}}_p,1)$ and $p_n(\widetilde{\mathbf{x}}) = N(\widetilde{\mathbf{x}}_n,1)$ be unit Gaussian distributions centered at $\widetilde{\mathbf{x}}_p,\widetilde{\mathbf{x}}_n$ respectively. We have

$$\mathbb{E}\left[-\mathcal{L}_{\text{triplet}}\right] \quad \propto \quad \mathbb{E}\left[\log p_p(\widetilde{\mathbf{x}}) - \log p_n(\widetilde{\mathbf{x}})\right] \tag{16}$$

$$\approx \quad \mathbb{E}\left[\log p_p(\widetilde{\mathbf{x}}) - \log p(\widetilde{\mathbf{x}})\right] \tag{17}$$

$$= \quad -H(\widetilde{X}|Y) + H(\widetilde{X}) = I(\widetilde{X};Y). \tag{18}$$

Two approximations were made in the process. We first assumed that the embedding distribution of images not in $\mathbf{y}_q$ is equal to the distribution of all embeddings. This is reasonable when each class only represents a small fraction of the full data. We also approximated the embedding distributions $p(\widetilde{\mathbf{x}}|\mathbf{y}), p(\widetilde{\mathbf{x}})$ with unit Gaussian distributions centered at single samples from each.

**N-pair Loss**    Multiclass $N$-pair loss (Sohn, 2016) was proposed as an alternative to Triplet loss. This loss function requires one positive embedding $\widetilde{\mathbf{x}}^+$ and multiple negative embeddings $\widetilde{\mathbf{x}}_1, \ldots, \widetilde{\mathbf{x}}_{N-1}$, and takes the form

$$-\log \frac{\exp(\widetilde{\mathbf{x}}^\top\widetilde{\mathbf{x}}^+)}{\exp(\widetilde{\mathbf{x}}^\top\widetilde{\mathbf{x}}^+) + \sum_{i=1}^{N-1}\exp(\widetilde{\mathbf{x}}^\top\widetilde{\mathbf{x}}_i)}. \tag{19}$$

This can be seen as the cross-entropy loss applied to $\text{softmax}(\widetilde{\mathbf{x}}^\top\widetilde{\mathbf{x}}^+, \widetilde{\mathbf{x}}^\top\widetilde{\mathbf{x}}_1, \ldots, \widetilde{\mathbf{x}}^\top\widetilde{\mathbf{x}}_{N-1})$.

Following the same logic as the cross-entropy loss, this is also an approximation to $I(\widetilde{X};Y)$. This objective should have less variance than Triplet loss since it approximates $p(\widetilde{\mathbf{x}})$ using more examples.

**Adversarial Metric Learning**   Deep Adversarial Metric Learning (Duan et al., 2018) tackles the problem of most negative exmples being uninformative by directly generating meaningful negative embeddings. This model employs a *generator* which takes as input the embeddings of anchor, positive, and negative images. The generator then outputs a "synthetic negative" embedding that is hard to distinguish from a positive embedding while being close to the negative embedding.

This can be seen as optimizing

$$\mathbb{E}\left[\log p_p(\widetilde{\mathbf{x}}) - \log p(\widetilde{\mathbf{x}})\right] = I(\widetilde{X}; Y) \tag{20}$$

by estimating $p(\widetilde{\mathbf{x}})$ using a generative network rather than directly from samples. Rather than modelling the marginal distribution $p(\widetilde{\mathbf{x}})$, this method conditionally models $p(\widetilde{\mathbf{x}}; \widetilde{\mathbf{x}}_q, \widetilde{\mathbf{x}}_p, \widetilde{\mathbf{x}}_n)$ so that $\widetilde{\mathbf{x}}$ is hard to distinguish from $\widetilde{\mathbf{x}}_p$ while sufficiently close to both $\widetilde{\mathbf{x}}_q$ and $\widetilde{\mathbf{x}}_n$.

## B   PROOF OF THEOREM 1

We restate and prove our main theorem.

**Theorem 1.** *Let $d_\Theta$ be the VC dimension of the encoder $\widetilde{X}(\cdot)$. Let $\hat{I}(\widetilde{X}(X_T, \theta); Y_T)$ be the empirical estimate of the mutual information using finite dataset $D_T$, and define empirical loss as*

$$\hat{\mathcal{L}}(T^{1:n}, \theta) = -\frac{1}{n}\sum_{i=1}^n \hat{I}(\widetilde{X}(X_{T^i}, \theta); Y_{T^i}). \tag{21}$$

*The following inequality holds with high probability:*

$$\mathcal{L}(\tau, \theta) - \hat{\mathcal{L}}(T^{1:n}, \theta) \leq O\left(\sqrt{\frac{d_\Theta}{n}\log\frac{n}{d_\Theta}}\right) + O\left(\frac{|\widetilde{X}|\log(m)}{\sqrt{m}}\right) + O\left(\frac{|\widetilde{X}||Y|}{m}\right) \tag{22}$$

*Proof.* We use the following lemma from Shamir et al. (2010), which we restate using our notation.

**Lemma 1.** *Let $\widetilde{X}$ be a random mapping of $X$. Let $D$ be a sample of size $m$ drawn from the joint probability distribution $p(X, Y)$. Denote the empirical mutual information observed from $D$ between $\widetilde{X}$ and $Y$ as $\hat{I}(\widetilde{X}; Y)$. For any $\delta \in (0, 1)$, the following holds with probability at least $1 - \delta$:*

$$|I(\widetilde{X}; Y) - \hat{I}(\widetilde{X}; Y)| \leq \frac{(3|\widetilde{X}| + 2)\log(m)\sqrt{\log(4/\delta)}}{\sqrt{2m}} + \frac{(|Y| + 1)(|\widetilde{X}| + 1) - 4}{m} \tag{23}$$

We simplify this and plug in our specific quantities of interest $(\widetilde{X}(X_T, \theta), Y_T)$:

$$\left|I\left(\widetilde{X}(X_T, \theta); Y_T\right) - \hat{I}\left(\widetilde{X}(X_T, \theta); Y_T\right)\right| \leq O\left(\frac{|\widetilde{X}|\log(m)}{\sqrt{m}}\right) + O\left(\frac{|\widetilde{X}||Y|}{m}\right). \tag{24}$$

We similarly bound the error caused by estimating $\mathcal{L}$ with a finite number of tasks sampled from $\tau$. Denote the finite sample estimate of $\mathcal{L}$ as

$$\hat{\mathcal{L}}(\tau, \theta) = -\frac{1}{n}\sum_{i=1}^n I(\widetilde{X}(X_{T^i}, \theta); Y_{T^i}). \tag{25}$$

Let the mapping $X \mapsto \widetilde{X}$ be parameterized by $\theta \in \Theta$ and let this model have VC dimension $d_\Theta$. Using $d_\Theta$, we can state that with high probability,

$$\left|\mathcal{L}(\tau, \theta) - \hat{\mathcal{L}}(\tau, \theta)\right| \leq O\left(\sqrt{\frac{d_\Theta}{n}\log\frac{n}{d_\Theta}}\right), \tag{26}$$

where $d_\Theta$ is the VC dimension of hypothesis class $\Theta$.

Combining equations (26, 24), we have with high probability

$$\left| \mathcal{L}(\tau, \theta) - \left( -\frac{1}{n} \sum_{i=1}^{n} \hat{I}(\widetilde{X}(X_{T^i}, \theta); Y_{T^i}) \right) \right| \tag{27}$$

$$\leq \left| \mathcal{L}(\tau, \theta) - \hat{\mathcal{L}}(\tau, \theta) \right| + O\left( \frac{|\widetilde{X}| \log(m)}{\sqrt{m}} \right) + O\left( \frac{|\widetilde{X}||Y|}{m} \right) \tag{28}$$

$$\leq O\left( \sqrt{\frac{d_\Theta}{n} \log \frac{n}{d_\Theta}} \right) + O\left( \frac{|\widetilde{X}| \log(m)}{\sqrt{m}} \right) + O\left( \frac{|\widetilde{X}||Y|}{m} \right) \tag{29}$$

$$\square$$

## C  EXPERIMENTS AND IMPLEMENTATION DETAILS

**Hardware**   Every experiment was conducted on a single Nvidia V100 GPU with CUDA 9.2. We used PyTorch version 1.0.1. Each experiment was performed with different fixed initial seeds; we manually fix seeds with `manual_seed()` for `python`, `pytorch`, and `numpy`.

**Optimizer**   For experiments with the 4-layer convnet, we use the `Adam` optimizer (Kingma & Ba, 2014) with learning rate 3e-4. For the Inception network, we use SGD with learning rate 3e-5 and momentum 0.9.

We report the average of 500 batches of 1-shot accuracies and mutual information. $I(\widetilde{X}; Y)$ was computed using balanced batches of 16 images each from 5 different classes. We additionally show in Figure 5 the correlation between 1-shot accuracies, Recall@1, and NMI using three previously proposed losses (triplet, npair, protonet).

**Small Train Set Experiment**   For this experiment, we used the Adam optimizer and performed a log-uniform hyperparameter sweep for learning rate $\in$ [1e-7, 1e-3] For DIMCO, we swept $p \in$ [32, 128] and $d \in$ [16, 32]. For other methods, we made the embedding dimension $\in$ [16, 32]. For each combination of loss and number of training examples per class, we ran the experiment 64 times and reported the mean and standard deviation of the top 5.

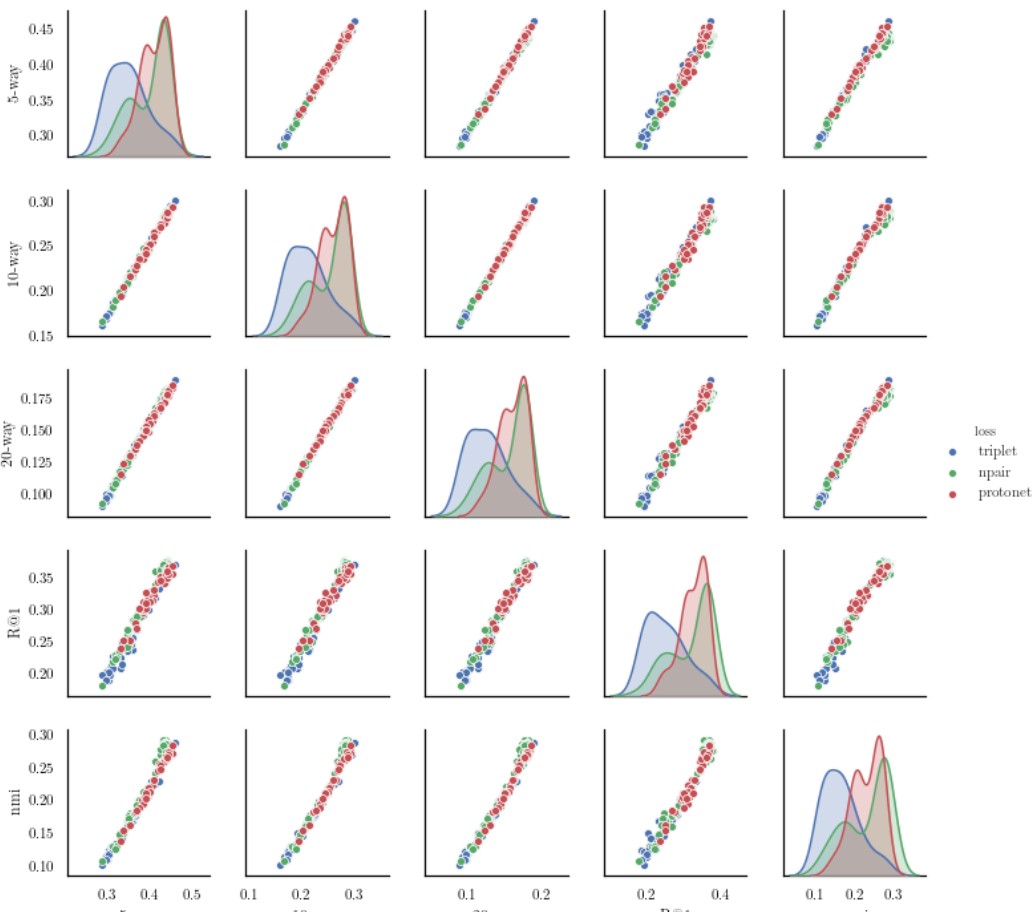

Figure 5: Correlation between few-shot accuracy and retrieval measures.

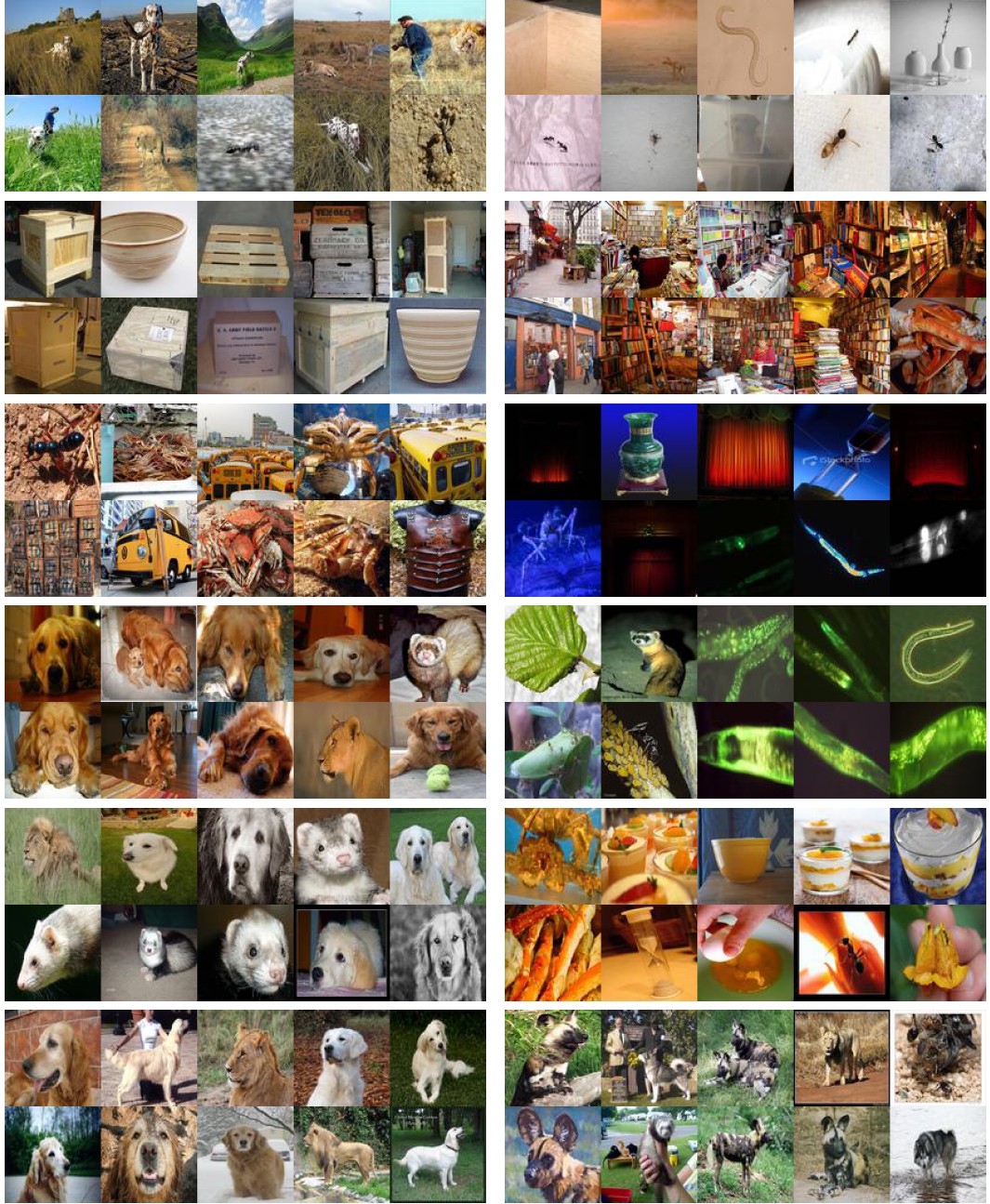

Figure 6: Additional visualizations of codes.

