# OpenReview forum: "Discrete InfoMax Codes for Meta-Learning"
_ICLR.cc/2020/Conference — Reject_

### Official Review · AnonReviewer2 · 2019-10-23
**Official Blind Review #2**

**Rating:** 3

**Review:**

This paper presents DIMCO, a meta-learner that is trained by maximizing mutual information between a discrete data representation and class labels across tasks. DIMCO is inspired by an information theoretic lower bound on the generalisation gap for meta-learning, which the authors argue identifies overfitting in the task learner as the bottleneck.

This work proposes to constrain a learner to output discrete codes that are learned to capture the mutual information with class labels. While the idea of using discrete codes is interesting, its presentation in the manuscript is not well motivated and at times hard to follow. This makes it challenging to evaluate the novelty, validity, and generality of the proposed approach. Meanwhile, the empirical evaluation is somewhat lacking. Thus, I do not believe this work is ready for publication in its current form.

Detailed comments

My main concern is with respect to the primary contribution of this paper, a generalisation bound on meta-learning. The bound appears to be on a multi-task loss without task adaptation, and thus the claims made with respect to the theorem seem somewhat over-reaching. I also believe the VC-dimensionality of the encoder is missing in Eq. 4? If so, this changes the interpretation since the length of the code and the expressivity of the encoder are interrelated. Further, I would welcome a deeper analysis of the theorem and its implications. The current interpretation states that the number of tasks is independent of the size of each task, hence given many tasks, using minimal representations is an effective approach to meta-generalisation. Yet minimal representations is a well-known idea and has features in several works that use mutual information as a regularizer, most notably works on the Information Bottleneck.

Another reservation I have is the use of mutual information between encoder representations and class labels as a loss function (Eq. 1). It lacks context and a proper motivation, especially since the analysis of [1] shows that the loss function in Eq. 1 is the cross-entropy objective. The authors make a similar analysis in Appendix A and argue that Eq. 1 differs in that cross-entropy is an approximation because it adds a parametrized linear layer on top of \tilde{X}. Thus, in the absence of that layer they collapse to the same objective. As DIMCO itself directly extract class label predictions from \tilde{X}, I fail to see a difference between the loss in Eq. 1 and a cross-entropy objective.

The main motivation behind their loss objective is that it does not require a support / query set. This does not seem to be a feature of the mutual information objective itself, but rather a choice made by the authors. I would have liked a deeper discussion of this seeing as the authors make it a central tenet of DIMCO. Prior works use a support set as a principled means of doing meta-learning: meta-training explicitly takes into account that at test time, the learner will be given a small support set from which to learn how to query points. As far as I understand, DIMCO does not take this into account during meta-training.  At meta-test time however, DIMCO does use a support set to map query points (Eqs. 10 and 11). Why should we break protocols between meta training and testing? Are there any downsides to doing so?

Empirically, I find the CUB experiment compelling but would welcome some ablations. What are the trade-offs between p and d? Can DIMCO outperform N-pair when number of bits are unconstrained?

miniImagenet is a standard benchmark in few-shot learning, but I am unable to find a results table - could the authors please provide results on the standard setup so that the method can be compared against known baselines? Further, would the results currently presented hold in a N-way-5-shot setup?

As for the constrained version of miniImagenet that the authors propose, I am not convinced this is an interesting protocol. In general, the miniImagenet task distribution is created by N-way permutations of the classes in the meta-set (e.g. meta-training tasks are combinations of the 64 classes in the meta-training set). By keeping the number of classes constant but reducing the number of images per class, this protocol is not reducing samples per task: a task is always defined as 5/20-way-1-shot (Fig. 4). Instead, the effect should be that tasks are in (greater) violation of the task i.i.d. assumption. Thus, I question whether this setup demonstrates the trade-offs the authors present in Theorem 1 and whether the results can be interpreted in light of it.

Finally, that both experiments are image-based raises questions as to the generality of the method. The paper could be considerably strengthened by evaluating DIMCO on a non-image task, or if not discuss the method’s limitations.

The idea of discrete codes for few-shot classification is interesting and sufficiently novel, I am likely to increase my score if my concerns are addressed and the experimental section is strengthened.

Further questions and comments:

- I am unable to parse Eq. 11 - what does the notation \prod_i p_{\tilde{x}_i, i} mean?
- It is unnecessarily hard to follow the proof of theorem 1. It would help the reader if you restated relevant definitions, such as Eq. 1, since the difference with Eq. 23 is very subtle. It would also be helpful to explain how the summand in Eq. 25 differs from either, and Eq. 26 could be expanded or briefly explained after the derivation.
-  Because DIMCO is trained with backpropagation, the interpretation of Eq. 5 as d independent events seems invalid. How does it affect the method if they are not independent?
- Overloading X and Y as both random variables and mini-batch samples creates unnecessary confusion. I believe the objective in Eq. 1 is approximated, not calculated exactly? For instance, the mutual information in Eq. 8 is with respect to a mini-batch, so should it not be \hat{I}?
- p^j_{ik} in Eq. 9 is undefined.
- Eq. 9 is interpreted as an exact entropy, however it appears to be a mini-batch approximation to the true entropy?

References
[1] Achille and Soatto. Emergence of Invariance and Disentanglement in Deep Representations. JMLR. 2018.

**Experience Assessment:**

I have published one or two papers in this area.

**Review Assessment: Checking Correctness Of Derivations And Theory:**

I carefully checked the derivations and theory.

**Review Assessment: Checking Correctness Of Experiments:**

I carefully checked the experiments.

**Review Assessment: Thoroughness In Paper Reading:**

I read the paper thoroughly.

---

> ### Author Response · Authors · 2019-11-15
> **Response to Review #2**
>
> “appears to be a multi-task loss without task adaptation”
> Our model does perform task adaptation, as one can see from the fact that it can classify new datasets given few labeled examples. The task adaptation happens implicitly, much like how there is no explicit “adaptation step” in the k-NN classification algorithm. Furthermore, we prove in the appendix that previous meta-learning and metric-learning setups (which do perform explicit task adaptation) can be seen as special cases of our problem setup.
>
> “VC-dim of the encoder is missing in eq 4? If so, this changes the interpretation since the length of the code and the expressivity of the encoder are interrelated.”
> Yes, there should be a VC-dim term, and there was one in the extended version of the theorem in Appendix B. We originally left it out of the main text to highlight the influence of n and m, but we now see that leaving it out causes more confusion than clarity. We have updated our draft to show the VC dimension term in the main text, as well.
> We agree that the encoder VC-dim and code length are interrelated. However, this does not change the interpretation since VC-dim only appears in the O(1/sqrt(n)) term, which we decided to ignore since n is so large. Even if we chose not to ignore that term, this relation is actually in favor of DIMCO’s approach since it suggests that we can decrease the VC-dim by decreasing codelength.
>
> “I would welcome a deeper analysis of the theorem and its implications. …minimal representations is a well-known idea…”
> Exactly. Minimal representations are already a well-established idea, and our paper’s main novelty is in figuring out exactly how that idea applies to the meta-learning problem. We have added some analysis to the generalization bound section, which we restate here.
> Meta-learning is typically formulated as performing two levels of learning: task-general learning and task-specific learning. Our theorem implies that in the tasks considered in recent literature, task-general learning should have almost no generalization gap, making task-specific learning the main source of (meta-)overfitting. This can be problematic since, in many works, the task-specific learning algorithm is usually just a byproduct of whatever clever meta-learning loss was proposed. Our theorem implies that we should pay more attention to directly regularizing this task-specific learner, and our DIMCO model can be seen as a minimal working example in this direction.
>
> “I fail to see the difference between (1) and a cross-entropy objective.”
> As you point out, [1] of your review and also Appendix A of our draft show that cross-entropy and MI are closely related. The issue is that cross-entropy cannot be used for our setup (meta-learning with discrete Xtilde). As you mentioned, we need some mapping of Xtilde->Y to calculate the cross-entropy, but having a fixed such map here is nonsensical since Y can be permuted while X and Xtilde remain the same. ProtoNets, for example, get around this problem by constructing this Xtilde->Y map for each new task using the mean Xtilde for each Y. This trick also cannot be used in our setup because we consider discrete Xtilde. In this light, we can view our MI objective as a convenient way to get around the difficulties in computing the cross-entropy directly.
>
> “Support/query set…does not seem to be a feature of MI objective itself, but rather a choice…why should we break protocols during meta-training and testing?”
> This is a feature of the MI objective rather than a choice we made since MI can only be computed for data that has both Xtilde and Y; there is no use for a “query set” in the MI loss. This is not a weakness since other meta-learning losses require the label corresponding to the query set anyway. While DIMCO could be viewed as having different protocols for training and testing, we argue that DIMCO learns using all the data provided, while other meta-learning methods are artificially hiding some of the labels in each meta-training set. We additionally point out that not splitting has the advantages of (1) being able to freely use batchnorm during meta-training and (2) being able to train even with datasets of one image per class (see last experiment of paper).
>
> “tradeoff btw p and d?”
> Please refer to our response to R1.
>
> “standard benchmarks?”
> The CUB experiment is meant to serve this purpose of showing DIMCO’s performance on a standard large-scale problem. As the specific form of DIMCO is closer to metric learning rather than e.g. MAML, we think that comparing on this metric learning benchmark sufficiently shows the large-scale applicability of DIMCO.
>
> “Can DIMCO outperform N-pair when number of bits are unconstrained?”
> Figure 3 shows that DIMCO does outperform N-pair loss. Our experience is that N-pair with d-dim continuous embeddings (for any reasonable d) is beat by DIMCO with (p, d)=(256, d).
>
> “…raises questions as to the generality of the method”
> Please refer to our response to R3.

---

### Official Review · AnonReviewer3 · 2019-10-28
**Official Blind Review #3**

**Rating:** 3

**Review:**

This paper proposes a method to learn classifier outputs for meta learning in the form of factorized discrete codes by maximizing the mutual information between the model's outputs and the ground truth labels. The authors further present an information theoretic generalization bound for meta learning in terms of the number of tasks, number of training samples per task and the expressively of the model. They further show empirically that their approach does not need a separate query set during meta-training and can generalize better than a few of the other metric-learning based meta-learning approaches specifically at lower shot values. They show that their method requires less memory than the N-pair meta-learning method.

The paper addresses an important problem of generalization with very low shot values and proposes an interesting theoretical treatment of the problem in terms of deriving an information theoretic lower bound for it. I liked the authors' theoretical treatment of relating various metrics to the mutual information between the models' outputs and their labels.

However, there have been many recent works attempting to address the problem of better generalization of meta learning models. Most notable among them is the work of Kim et al. "Bayesian Model-Agnostic Meta-Learning" (https://arxiv.org/pdf/1806.03836.pdf), which is not referenced or compared against in this paper at all.

The proposed method is also limited in its scope to classification tasks only and the authors make no attempt to address regression or reinforcement learning problem, which limits is widespread applicability.

While the authors address the generalization of meta-learning methods for small values of M, they do not address how their model behaves viz-a-viz others when the number of tasks N available for training is small. Theoretically having more compact representations should also help in situations where the number of tasks available for training are small. I would like to see an empirical analysis of that as well.



**Experience Assessment:**

I have published one or two papers in this area.

**Review Assessment: Checking Correctness Of Derivations And Theory:**

I assessed the sensibility of the derivations and theory.

**Review Assessment: Checking Correctness Of Experiments:**

I carefully checked the experiments.

**Review Assessment: Thoroughness In Paper Reading:**

I read the paper at least twice and used my best judgement in assessing the paper.

---

> ### Author Response · Authors · 2019-11-15
> **Response to Review #3**
>
> “…many recent works attempting to address the problem of better generalization in meta-learning…”
> We agree that BMAML tackles a similar problem, and we have added a discussion of this work to the related works section.
>
> “The proposed method is also limited in its scope to classification tasks only”
> This is true, and we have updated our manuscript to be more clear of this limitation. However, our paper is not necessarily unrelated to other problems. We connect the well-established information bottleneck concept to our core message of “to regularize meta-learners, we should restrict the task-specific learner” which can be used as a guiding principle for the design of any meta-learning method. We point out that previous methods such as MT-Net, LEO, and CAVIA have benefitted from precisely this restriction of only the task-specific learner. All three methods can be applied to both regression and RL.

---

### Official Review · AnonReviewer1 · 2019-11-06
**Official Blind Review #1**

**Rating:** 3

**Review:**

The paper proposes to maximize improve generalization in meta-learning by learning discrete codes via a mutual information maximization objective. I liked the motivation and presentation of the paper but see some critical shortcomings:

- In Theorem 1, shouldn't there be a term that accounts for the complexity of the hypothesis class (VC-dim, Rademacher complexity etc.). Can we actually verify Theorem 1 in practice on synthetic distributions to get a sense of the constant terms?
- The experiments do not compare with any mutual information related baselines. Eg, VIB [Alemi et al.]. This comparison is critical to stress the importance of discrete codes as opposed to continuous.
- Can the authors shed more light on the tradeoffs between p and d? Empirical insights would lend more intuition and understanding.
- Discreteness in activations is one form of regularization to reduce |tilde(x)|. Would regularizing the weights of the stochastic encoder (say by variational inference or minimizing say l2 norm) have the same/better/worse regularization effect (as done in Bayes by Backprop)?
- There are a few missing references on stochastic encoders trained based on variational information maximization [1, 2].

References:
[1] Uncertainty Autoencoders: Learning Compressed Representations via Variational Information Maximization. AISTATS 2019.
[2] Neural Joint Source-Channel Coding. ICML 2019.

**Experience Assessment:**

I have published one or two papers in this area.

**Review Assessment: Checking Correctness Of Derivations And Theory:**

I assessed the sensibility of the derivations and theory.

**Review Assessment: Checking Correctness Of Experiments:**

I carefully checked the experiments.

**Review Assessment: Thoroughness In Paper Reading:**

I read the paper at least twice and used my best judgement in assessing the paper.

---

> ### Author Response · Authors · 2019-11-15
> **Response to Review #1**
>
> “In Theorem 1, shouldn’t there be a term that accounts for the complexity of the hypothesis class?”
> There should, and there was such a VC-dimension term in the extended version of the theorem in Appendix B. We originally left it out of the paper to highlight the influence of n and m, but we now see that leaving it out causes more confusion than clarity. We have updated our draft to show the VC dimension term in the main text, as well.
>
> “Can the authors shed more light on the tradeoff btw p and d?”
> p and d jointly determine the codelength through the equality |Xtilde| = p^d. For example, (p, d)=(10, 5)=(100000, 1) have the same code length, but the latter option requires much more memory since the last layer must output a pd-dimensional vector. The tradeoff is that we want p^d to be large enough for expressivity, but also want pd to be small enough for memory constraints.
>
> “The experiments do not compare with MI baselines (e.g., VIB)”
> Thank you for raising this important point: there is no apparent way to use such continuous MI estimators (MINE, VIB, DIM, CPC, etc.) in the meta-learning setup. The core obstacle is that such methods require one neural network for each pair of r.v.s to estimate the MI of, but every task is a new pair of r.v.s in the meta-learning setup. This would mean that we use a different neural network just for estimating the MI within each task: this scheme does not generalize to new tasks (which is the point of meta-learning). By considering discrete MI, our loss can estimate both MI and predictions (I(Xtilde; Y) and p(Y|Xtilde)) in closed form. To summarize, the use of discrete codes was necessary for adapting the “maximize MI” idea to meta-learning.
>
> “Discreteness in activations is one form of regularization to reduce |tilde(x)|. Would regularizing the weights of the stochastic encoder (say by variational inference or minimizing say l2 norm) have the same/better/worse regularization effect (as done in Bayes by Backprop)?”
> Regularizing meta-learners is relatively new territory, and we believe that such problems should be investigated starting from the simplest solution conceivable. This is why we constructed a method that chooses from a finite hypothesis space. While we do not have enough information to answer your question directly, we believe that such alternative means of restricting task-specific learning capacity are an exciting research direction.
>
> “missing references”
> Thanks for the pointers to relevant works; we have added discussions of them to the related works section.

---

### Author Response · Authors · 2019-11-15
**Overall Response to Reviews**

Dear Reviewers,

Thank you all for your constructive feedback. Based on your comments, we have updated our manuscript to:
1. more precisely state our main theorem
2. state the limitations of our model
3. include relevant work that we missed.
We individually respond to your specific comments below.

---

### Decision · Program_Chairs · 2019-12-19

**Decision:**

Reject

**Comment:**

The reviewers were unanimous that this submission is not ready for publication at ICLR in its current form.

Concerns raised included that the method was not sufficiently general, including in choice of experiments reported, and the lack of discussion of some lines of significantly related work.